# Foraging distance distributions reveal how honeybee waggle dance recruitment varies with landscape
Joseph Palmer[1,2], Ash E. Samuelson[1], Richard J. Gill[3], Ellouise Leadbeater [1,4] &
Vincent A. A. Jansen [1] ✉

Honeybee (*Apis mellifera*) colonies use a unique collective foraging system, the waggle dance, to communicate and process the location of resources. Here, we present a means to quantify the effect of recruitment on colony forager allocation across the landscape by simply observing the waggle dance on the dancefloor. We show first, through a theoretical model, that recruitment leaves a characteristic imprint on the distance distribution of foraging sites that a colony visits, which varies according to the proportion of trips driven by individual search. Next, we fit this model to the real-world empirical distance distribution of forage sites visited by 20 honeybee colonies in urban and rural landscapes across South East England, obtained via dance decoding. We show that there is considerable variation in the use of dancing information in colony foraging, particularly in agri-rural landscapes. In our dataset, reliance on dancing increases as arable land gives way to built-up areas, suggesting that dancing may have the greatest impact on colony foraging in the complex and heterogeneous landscapes of forage-rich urban areas. Our model provides a tool to assess the relevance of this extraordinary behaviour across modern anthropogenic landscape types.

In group living animals, collective decisions can be taken by integrating information from multiple individuals to produce behaviour that extends beyond that of the individual[1]. Collective behaviour thus emerges from simple behavioural 'rules' which filter social information[2,3]. In honeybees, and other eusocial insects, the behavioural architectures that produce emergent behaviours have become particularly complex. Honeybee colony foraging is coordinated via the waggle dances of individual foragers, which communicate food source locations to nestmates (Fig. 1). A series of behavioural rules that determine when, and how much, bees dance mean that choices between feeding sites can be made by the group[4–6]. For example, the number of dance circuits performed by a forager on returning from a food source reflects the net energetic benefits of the trip[6]. As a result, more of the colony's workforce will be recruited to the closer of two equally rich sources[7] or the richer of two equidistant sources[5]. This filtered recruitment mechanism allows a colony to allocate collective foraging effort, effectively choosing which of the available forage sites to focus upon, without the need for any individual to compare resources.

Dancing is a universal feature of honeybee behaviour and is commonly observed in all *Apis mellifera* colonies. However, individual bees only respond to the spatial directions provided by dances under particular circumstances[8,9].

For example, foragers that have current knowledge of a resource site are rarely influenced by dances for alternative sites[10], even if those alternatives are more rewarding[11]. In contrast, dances are highly influential for foragers whose known sites become depleted[8,12,13], or after temporal gaps in foraging. As a result, the importance of dance communication—and therefore collective decision-making—for a colony's choice of forage sites should depend on resource distribution in the landscape. Environments where recruitment is influential are intriguing, because they are likely to have been important in driving the evolution of the waggle dance, yet empirical attempts to identify them have produced mixed results[14–21]. Initial work, in which dances were rendered meaningless by preventing bees from referencing the sun's position (Fig. 1), tentatively linked the benefits of collective foraging to landscape characteristics such as heterogeneity[21]. However, empirical attempts to systematically test this relationship have failed to provide support[14,15], and dance disruption has sometimes even been associated with higher, rather than lower, foraging success[19]. Consequently, no clear pattern has yet emerged with respect to the ecological conditions that determine whether colonies forage collectively, or as a group of individuals[22].

Here, we present a methodology to infer the influence of recruitment on colony foraging, and to quantify differences in this influence across

[1]Department of Biological Sciences, Royal Holloway University of London, Egham, Surrey, TW20 0EX, UK. [2]The Alan Turing Institute, 96 Euston Road, London, NW1 2DB, UK. [3]Department of Life Sciences, Imperial College London, Silwood Park Campus, London, UK. [4]Present address: Department of Genetics, Evolution and Environment, University College London, London, WC1E 6BT, UK. ✉e-mail: vincent.jansen@rhul.ac.uk

**Fig. 1 | The honeybee waggle dance carries information about the location of a resource. a** The honeybee waggle dance communicates the direction of the resource, relative to the direction of the sun through the angle of the dance relative to the vertical. The duration of the waggle run indicates the distance to the resource. **b** This information, in the form of a bearing and a distance, allows other foragers to locate the resource in the landscape (blue circle).

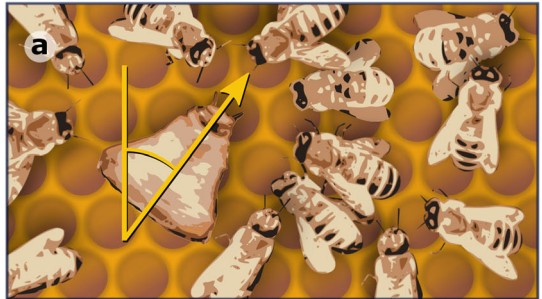
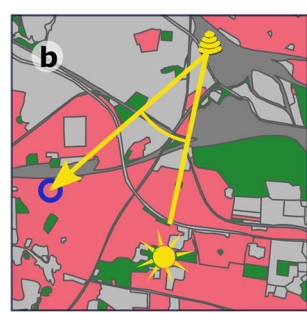

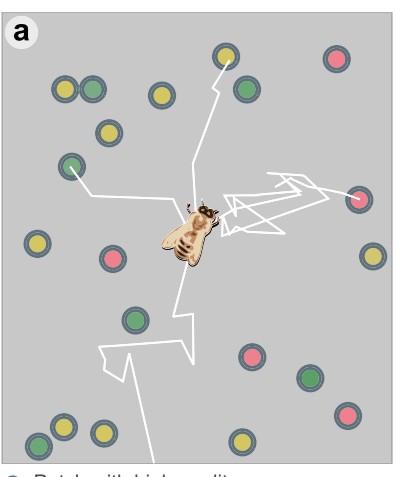
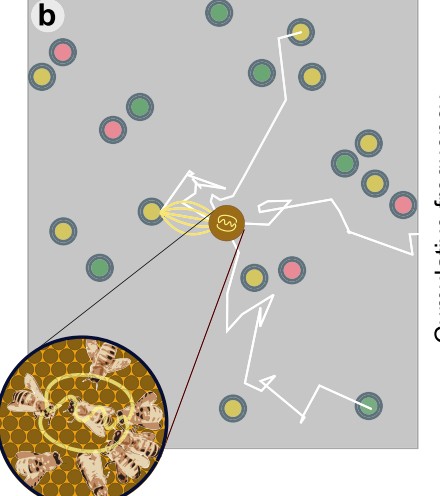
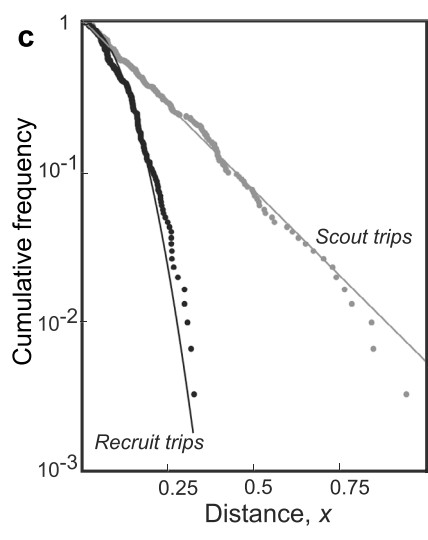

- ● Patch with high quality resource
- ● Patch with medium quality resource
- ● Patch with low quality resource

**Fig. 2 | Honeybee foraging. a** A hypothetical illustration of a typical foraging honeybee, foraging with scouting only. Foragers leave the hive on a search path with straight segments (white lines) and continue until they encounter a resource (coloured circles). **b** An illustration of honeybees foraging with recruitment. Foragers that have identified resources in scouting trips (white lines) convey this information on the dance floor (brown disc), where foragers can sample dances reporting on both scouting and recruiting trips and follow these directions (yellow lines). **c** Complementary cumulative frequencies of foraging distances reported from scouting and recruit trips from a simulation model (see Materials and Methods). Note the difference in the shape of the distributions. The scout distribution is best fit by an exponential (grey straight line), the recruit distribution by a Rayleigh distribution (black curve).

landscape types, simply by observing the dances on a colony's dancefloor. Using this method, we will show that the importance of dancing for colony foraging patterns is typically high in forage-rich, complex urban areas, but more variable at rural sites that are dominated by intensive agriculture. We first develop a theoretical model to establish how colony foraging patterns should appear when recruitment is used to varying degrees, compared to cases where all bees search for food individually. We find that recruitment leaves a characteristic "humped" imprint (see Results section "Recruitment creates a characteristic pattern in the distribution of foraging distances") on the cumulative distribution of distances reported on the dance floor, the magnitude of which correlates with the use of the waggle dance for collective foraging. We then fit these theoretical distributions to an empirical data set consisting of observations of waggle dances from 20 real-world hives in two different landscape types—urban and agri-rural—to quantify the relative contribution of waggle dance recruitment to colony foraging decisions in each case. Finally, we relate the variation in waggle dance use that we identify to local land-use patterns.

## Results
### The distribution of foraging site distances differs between scout and recruit trips
To establish how collective foraging influences the distribution of the distances advertised on the dance floor, we first consider the distribution of

food sites that would be located by bees following either a "scout" or a "recruit" strategy. When behaving as a scout, a bee seeks out resources by herself (Fig. 2a), and when behaving as a recruit she selects a random dance and locates the advertised resource (Fig. 2b)[6]. Note that scouts and recruits are not fixed behavioural categories, but rather, strategies adopted.

Since scouts often leave the hive on a straight search path and continue until they encounter a resource, the distances of the resources that they find should be distributed exponentially, following the nearest neighbour distance for foragers operating in a one-dimensional environment. We confirmed that these scout trip distances are captured by an exponential distribution through a simulation model (Materials and Methods) in which bees leave a central hive and search the surrounding two-dimensional space for randomly distributed resources (Fig. 2). Scouts communicate these distances on the dance floor, but sites are not represented equally. Instead, sites offering higher net energetic gains are over-represented on the dancefloor through the performance of more dance runs[6], so recruitment is more likely for closer sites[7,23], assuming that resource quality does not follow any systematic pattern with distance from the hive. Thus, the foragers that sample dances—the recruits—are biased towards the more profitable, closer resources. The distances of recruiting trips are then distributed through a Rayleigh distribution (again, confirmed through our simulation model; Fig. 2), which is the distribution one would expect if bees can intensively search the two-dimensional environment around the hive and select the

**Fig. 3 | The rationale underlying the derivation of foraging distributions. a** The distance distribution of resources encountered by scouts is approximated by an exponential distribution. **b** These distances are communicated on the dance floor. **c** Dances for resources that are nearer or higher in quality are repeated more often. The most profitable resources (red circles) are danced for more often for a given distance than resources of lesser quality (yellow and green circles). **d** Scout trip distances are translated into dance duration. **e** Recruits select a resource location from the dance floor (brown disc) and visit the resource (yellow disc). **f** The distance distribution of resources encountered on recruit trips is translated into dance duration. **g** By taking together the dance duration distributions for scouts and recruits the distributions of all dances on the dancefloor can be found, leaving a distinctive imprint in the form of a shoulder, or hump (**h**), which also leaves a similar imprint in the cumulative distance distribution of resources visited (**i**). The dots in the panels **a**, **d**, **f**, **g**, **h** and **i** are from simulations and are for illustrative purposes only.

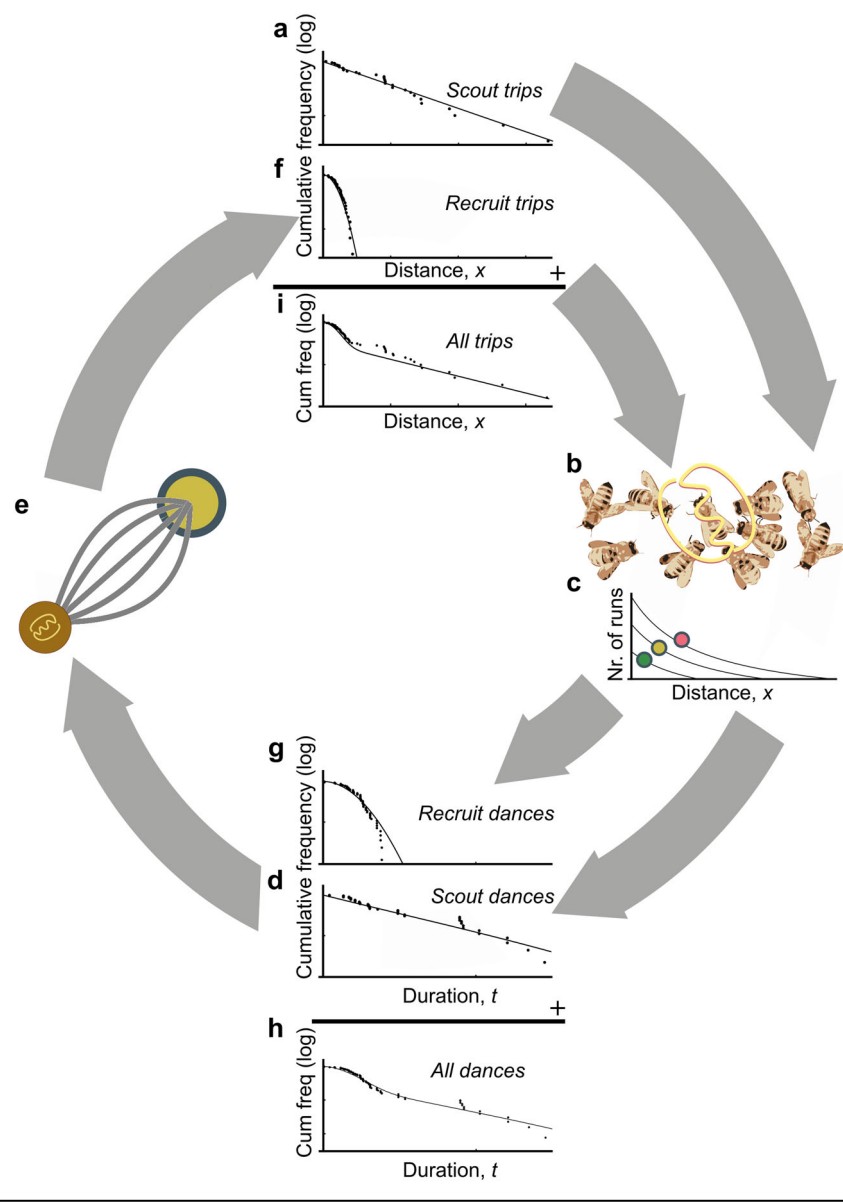

nearest resource. After successfully visiting advertised resources, recruits also dance for them, leading to further amplification of this bias towards the most profitable resource in the vicinity of the hive.

## Recruitment creates a characteristic pattern in the distribution of foraging distances

To predict the distribution of foraging distances that should be observed across whole colonies, we combined the scout and recruit distributions proportions in a mathematical model (Fig. 3; Material and Methods), where the proportion of scouts ($p$) could vary. Note that the lower the value of $p$, the higher the number of trips that are driven by recruitment, and thus the more important the role of dance communication for colony foraging. A mixture of scout and recruit trips results in an imprint in the form of a shoulder or "hump" for shorter distances in the cumulative distribution (Fig. 3h, i). The more recruitment takes place, the more pronounced this hump becomes. The full characterisation of the model can be found in Materials and Methods, Eqs. (1)–(3).

## The model captures real-world foraging distance distributions

To infer the use of communication and collective foraging in honeybee colonies foraging in 'natural' landscapes, we fitted our model to empirical data from a pre-existing dataset[24] of 2827 decoded waggle dances from 20

observation hives at different locations (10 in urban sites, 10 in agri-rural areas) in South East England (Fig. 4a), where dances had been recorded every fortnight from April to September 2017 (see Materials and Methods).

For each hive, we explored whether foraging was best described as an individual or collective venture. To do so we fitted two versions of our model using maximum likelihood methods. The first was an "individual" model, where all forage sites are found through scouting, and the second was a "collective" model, where the proportion of scout trips ($p$) could take on any value between 0 and 1. For each hive, we used the Akaike Information Criterion (AIC) to determine which model version provided the better explanation of the data[25]. If the collective model provided a better fit, we quantified the relative importance of waggle dance communication through estimating the parameter $p$. We also calculated the goodness-of-fit using a Kolmogorov–Smirnov (KS) test to ascertain if the models (individual and collective) provided a plausible explanation of the data[26,27].

The hives shown in Fig. 4c, d are representative examples showing the model fits where the individual (Fig. 4c) and collective (Fig. 4d) models fit best; for the full set see Supplementary Fig. 1. Note the closeness of the fit to the data, illustrating the overall quality of the model description.

In four cases (3 agri-rural, 1 urban), model selection identified the individual model as having a better fit to the data than the collective model ($1.83 < \Delta AIC < 4.32$; KS test $P$-value $> 0.05$ for all collective and individual

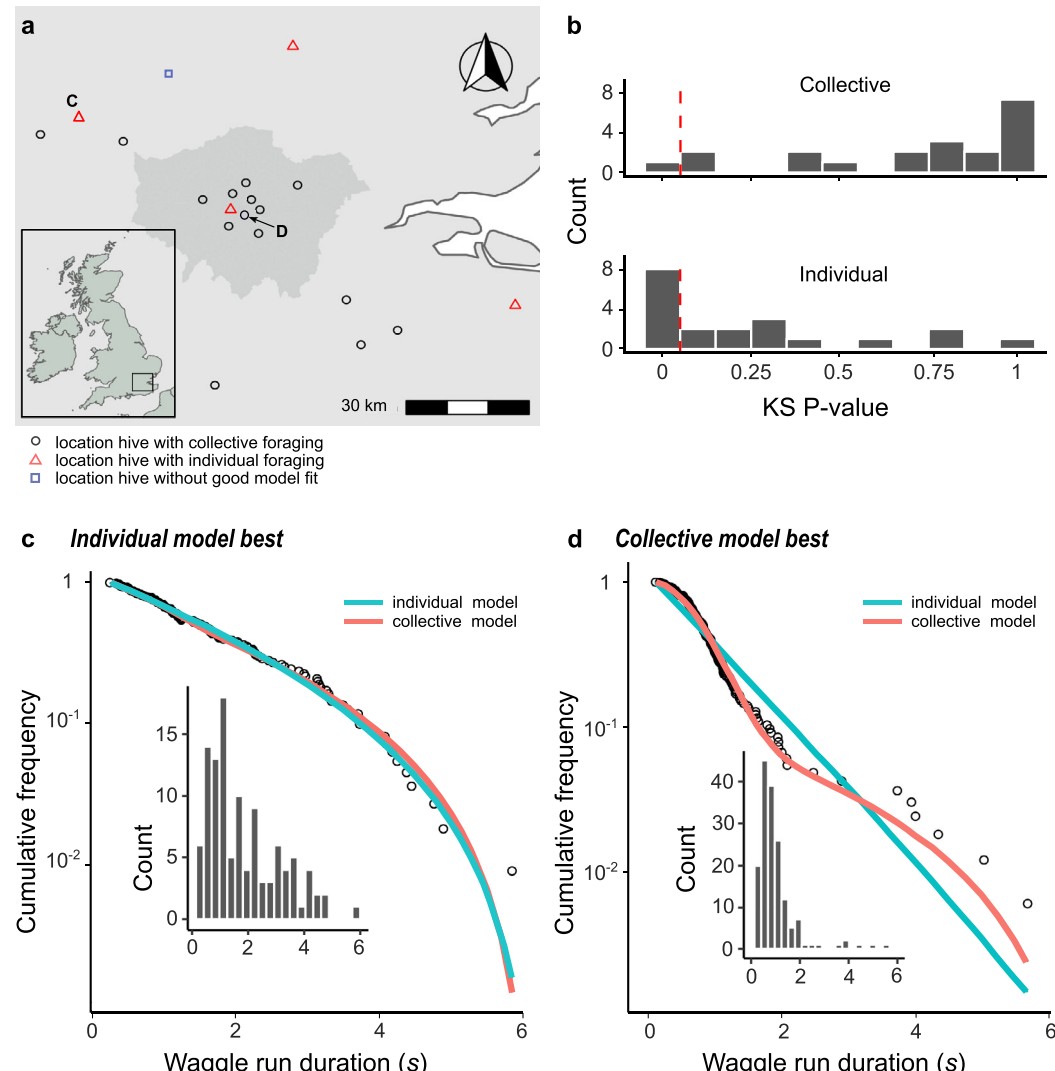

**Fig. 4 | The honeybee foraging model fitted to data from 20 hives. a** Location of study hives in South East England, shaded area in the main plot indicates Greater London. For 16 hives the collective foraging model provided best explanation (circles), for 4 hives the individual search model provided the best explanation (triangles) as indicated by lowest AIC score, and for one hive (square) neither model gave a good fit. Locations of hives that provided data for subsequent panels indicated by **c** and **d**. **b** Distribution of goodness of fit confidence values for each model fit to waggle run durations from each site. The $P$-value is derived from a bootstrapped two-sided KS test comparing the fitted model predictions to the empirical data, the red dashed line marks the significance threshold of 0.05. The number of data points used for each site is given in Supplementary Table 1. **c** Representative example of the dance duration distribution for a hive in which the individual model (blue line) provided a better fit than the collective foraging model (red line). **d** Dance duration distribution for a hive in which the collective foraging model (red line) provided a better fit than the individual model (blue line). The typical shoulder or "hump" in the distribution is indicative of contribution of recruitment dances. Panels show the complimentary cumulative frequencies with binned frequency distributions as inset.

models). For the 16 remaining hives, the collective model provided a better fit ($2.07 < \Delta AIC < 49.43$), and in 7 of these cases (2 agri-rural, 5 urban), the collective model was the only one to provide an acceptable fit to the observed data (KS test $P$-value > 0.05). For one agricultural hive, neither model had an acceptable fit (KS test $P$-value < 0.05 for both collective and individual models). In summary, 15 hives showed a detectable imprint of collective foraging, while 4 were best described by a scenario that involves no recruitment, and one was not well captured by either model.

These results indicate that, whilst colony-level foraging is mostly comprised of a mixture of scout and recruit foraging trips, in some circumstances, it can be better described by individual foraging alone. Thus, in some environments, most foraging trips involve sites found through individual search rather than dance recruitment. Note that this does not imply that these bees do not engage in dance following, because bees regularly follow dances but choose not to visit the advertised site[28], and we do not expect individuals to follow different rules at different sites. Rather, our results suggest that the circumstances under which bees respond to dances by seeking out the resources they advertise—e.g. when their current food source has depleted and alternatives are being advertised—are more common at some sites than others.

## Waggle dance use reflects landscape structure
Our approach is not limited to a binary explanation of colony-level foraging as collective or independent. Quantifying the influence of waggle-dance recruitment on colony foraging patterns, as a proportion of all foraging trips, can be achieved by estimating the proportion of recruit trips, $1\text{-}p$, henceforth termed "waggle dance use". Our dataset pre-dated this study and was not specifically designed for the analysis performed here; the hives from which data were collected represented different landscape types because they were in either urban or agri-rural locations. In urban areas in SE England, forage for social bees is typically relatively abundant, diverse and consistent across the season, while agri-rural sites are thought to be characterised by long

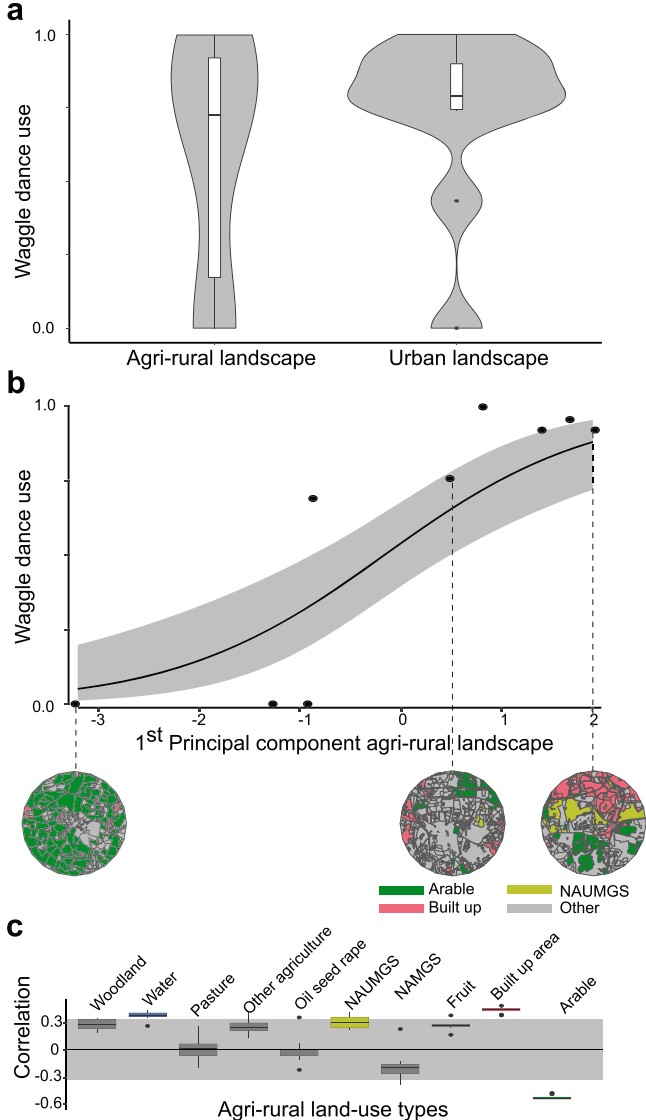

**Fig. 5 | Collective foraging correlates with land-use. a** Within both the agri-rural and urban landscapes, we found considerable variation in waggle dance use across sites. **b** We thus sought to explore whether this could be explained by differences in land-use across sites using PLS analysis. For the agri-rural landscape beta regression shows the relationship (black line) between first principal component and waggle dance use, with 95% confidence interval shown by the grey shaded area.
**c** Correlation scores for the PLS. Boxplots illustrate the range of correlations (median, upper and lower quartiles) between the first principal component and each land-use type following jack-knife resampling. Correlations outside the shaded area significantly contribute to the first principal component. Going from left to right in panel **b**, the landscape will contain fewer landscape types with a low correlation score and more with a higher correlation score. NAUMGS and NAMGS stands for non-agricultural unmanaged green space and non-agricultural managed green space, respectively.

periods of relatively sparse forage interspersed with temporary high abundance through mass-flowering crops[29]. We investigated how waggle dance use varied across these landscape categories.

The violin plots in Fig. 5a show that while waggle dance use was often high for both the urban and agri-rural landscapes, the agri-rural sites showed considerably more variation than the urban sites. Since agri-rural sites vary in many respects, including the amount of localized urbanization, pasture/arable land, and other factors that will influence food availability for bees, we further explored whether aspects of land-use within the agri-rural category correlated with waggle dance use. We performed a GIS-based land-

use classification to obtain a quantitative land-use profile for each of the agri-rural sites[30] based on the combination of land-use types present (e.g. woodland; arable land; pasture; fruit crops), followed by a Partial Least Squares (PLS) analysis[31] to identify the combinations of land-use types which best explained the variation in our estimates of dance use. We found that a single principal component that correlated negatively with arable land cover (29% of total land area; Supplementary Table 2) and positively with built-up area (17% of land area; Supplementary Table 2) explained ~73% of the variation in waggle dance use (beta regression: $R^2 = 0.73$, $\phi = 4.9$, $P < 0.05$, Fig. 5). Waggle dance use tends to increase if these factors become more dominant in the landscape (Fig. 5b; note the decreases in the proportion of arable land and increase in built up area in the map insets in the figure from left to right). This principal component also correlated positively with non-agricultural unmanaged green space (e.g. set-aside land, scrub), though this relationship was not robust to jack-knife resampling (Supplementary Fig. 2). This suggests that forage was more likely to be found through recruitment in those agri-rural sites that were relatively built-up (e.g. close to towns and with more gardens), and individual search dominated at the more arable, genuinely rural sites. This also potentially explains why the variation in the impact of dance communication in urban landscapes is lower, since the proportion of built-up area, and with that forage-rich gardens, is generally higher. The one urban site that was better described by the individual model than the collective was also the site with the lowest proportion of developed land within the urban category (66%, relative to a range of 72–91% at other urban sites), due to a large local water body.

## Discussion

Our results show that there is considerable variation in the impact that waggle dance recruitment has on the distribution of forage sites chosen by honeybee colonies foraging naturally across anthropogenic landscapes[8,9]. We do this by using a purely observational approach that does not disrupt the natural foraging behaviour of the colony: by simply decoding the dances produced by a colony's workforce and fitting our model to the foraging site distances obtained as a result, it is possible to see that the foraging patterns of some colonies bear a clear hallmark of recruitment, whereas others do not.

What causes this variation? Our metric for assessing the impact of dance recruitment is the proportion of bees that follow a "recruit" strategy. Bees that use this strategy sample a dance from the dance floor before leaving the hive to find the site communicated by the dance, rather than searching independently. At the proximate level, the factors that underlie the choice to act as scout or recruit are not well understood, although previous work has found that certain gene expression profiles are predictive of scouting behaviour[32] and that the tendency to act as a recruit is greater in younger bees[13,33]. But it is clear that these strategies are sufficiently flexible to allow changes in the proportion of scouts with local foraging conditions, and Seeley[33] observed through intensively tracking individual bees in observation hives that the proportion of scouts decreases dramatically with forage availability, as the number of dances increases[33]. Thus, there is reason to expect that the hives we identified as relying heavily on dance recruitment are those that are in forage-rich areas.

Accordingly, we found patterns in the importance of waggle dance recruitment that reflected landscape structure, such that variation was higher in agri-rural than urban environments, and within the agri-rural category, increased along an axis that reflected a transition from forage-poor agricultural land towards increasing residential development. Agricultural land in the UK is typically considered nutritionally poor for bees, with large areas of limited food availability punctuated by brief availability of rich mass-flowering crops in some areas, while more urbanized residential areas that contain gardens are relatively forage rich, with many diverse small patches of flowers in residential gardens and allotments[29,34,35]. We speculate that the dancefloors of hives in these urbanized areas may contain dances advertising multiple alternative sites, such that when rewards decrease at one site, dances advertising others are quickly encountered. At sparser agri-rural sites dominated by arable land, where food is potentially more challenging to find, representation of

multiple alternatives on the dancefloor may be rarer. Our current dataset lacks the temporal resolution to explore whether recruitment is more common at times when greater diversity is likely within such landscapes, such as late spring, but further work could focus on rural locations systematically chosen to represent a range of floral diversity and abundance (e.g. ref. 36), with the temporal resolution to focus on specific periods of the year, to identify those ecological contexts in which dance communication has a detectable impact on colony foraging.

It is widely assumed that certain aspects of the tropical forests in which *Apis* evolved favoured the evolution of the dance[22]. Current attempts to identify those critical features rely upon elegant but labour-intensive dance-disorientation protocols, whereby dance communication is actively prevented and the consequences for foraging efficiency monitored[16,17,21,36,37]. Our method complements and builds upon these studies. On the one hand, our findings contribute to explaining why dance disorientation sometimes has little effect on foraging efficiency in some environments[14,15,17,19–21]: although dancing takes place, recruitment is relatively rare. Rather than collective foraging being an inflexible adaptation to a landscape type, this suggests that the feedback loops that regulate collective foraging also provide the mechanism to fine-tune the foraging to fine-grained spatial or temporal variation in food availability. On the other, this method has the potential for expansion across multiple landscapes and land-use types. We have described a means to quantify the extent to which honeybee colonies rely on recruitment that functions simply by observing and decoding the dances on the dancefloor, and although we were limited by the need to manually decode dances, further study will not be. Given the recent development of automated dance-decoding protocols[38,39], now also validated for field-based videos[40], our inferential method has broad-ranging potential to provide a time and labour efficient means to identify the environments in which this iconic example of animal communication shapes collective behaviour.

## Materials and methods
### Overview
To predict the distribution of forage site distances reported by bees employing individual search (scouts) or following dances (recruits), we built a theoretical model to describe the distribution of distances between the hive and forage sites for any honeybee colony. This model contained "scout" trips- described by an exponential distribution, and "recruit" trips, described by a Rayleigh distribution. Both distributions were validated during model development by means of a simulation, described below. Using a pre-existing dataset of foraging distributions extracted from videos of waggle dances in 20 hives, for each hive we compared the fit of this collective model with a model in which foraging is entirely independent. Finally, for each hive within the agri-rural landscape ($n = 10$ hives), we calculated a multivariate land-use profile to describe the surrounding landscape and performed a Partial Least Squares analysis to establish how the proportion of bees acting as recruits ("waggle dance use", estimated from the collective model) varied with land use.

### Simulation
A circular foraging environment was created with radius $r = 2.5$. The number of resource patches in the environment was generated as a random Poisson variable with a mean of 1/5000 multiplied by the area of the environment. Resource patches are located at polar coordinates with a uniformly selected angle, $\theta$, between 0 and $2\pi$ and a radial value, $\rho$, between 0 and $r$, determined by the square root of the uniform position values multiplied by $r$. Each location was assigned to an instance of a resource object, where resource quality was randomly allocated between 0 and 10. Resource profitability is a product of this quality and the distance of the resource to the centrally located hive. Resources were periodically replaced with new resource of a random quality, on average once or twice per simulation.

One hundred honeybee objects forage in this environment. Foraging bees follow independent flight segments with random lengths and direction[41]. In our simulation scouts follow a random path through the environment generated by sampling a uniform random step length and angle. The number of paths the scout draws when searching is also determined as a uniform random number. Each straight line in the random path is converted to a rectangle with length equal to the path section length and a constant width of 0.01 to represent an area the scout searches along that path. Of all the resources contained in the boxes drawn from the scout's path, the one closest to the colony is selected as the resource patch that the scout will report if the quality of the resource exceeds a minimum threshold. Communication is simulated by pooling all the resource patches found. If no resources are contained in the scout's path, no resources will be added to the scout pool.

Recruits are honeybees that do not perform individual searches for forage, but instead sample from the pool of resource objects reported by scouts. The probability of sampling a resource is skewed by its profitability, mimicking the profitability bias known to occur in the dances of real-world scouts[6]. Recruits then visit these resources, and in the next iteration will add their resource to the pool of scout dances. Consequently, the pool of dances represents resources discovered by scouts and resources exploited by recruits. When a resource is depleted, it is removed from the environment and so any foragers that had been visiting it must select a different resource from the dance floor.

The simulation was run for 100 iterations, in which all distances reported by scouts and recruits were recorded and combined every 5 time steps. We fitted an exponential and minimum of an exponential distribution to the distribution of foraging distances reported by the (Fig. 1a) scout and (Fig. 1b) recruit objects, by deriving the maximum likelihood estimate for each model fit on each data source through their analytical solutions: $\hat{\lambda} = 1/\bar{x}$, minimum of the exponential with a minimum foraging distance: $\hat{\lambda} = 1/(\pi \bar{x}^2)$. As the exponential assumes that distributions start from 0, the data were transformed by subtracting the minimum foraging distance from all foraging distances ($x = x - \min(x)$) before fitting. All simulation code was written in Python version 3.9 and uses the pandas[42] and SciPy[43] packages.

### Theoretical model
The duration of the waggle run of a dance circuit represents the distance flown by the bee, and the two are linearly related[44,45]. To describe the distribution of waggle run durations on the dance floor we formulated a generic statistical model for the duration of waggle dances, in which it is assumed that resource patches are randomly placed in the environment. Foragers scout for these patches. Upon visiting a resource patch, foragers translate the profitability of a resource into the number of repeats of the dance, also called "dance circuits" (Fig. 2). The number of dance circuits is a function of quality and distance. Recruits sample random dances and report the location of successful visits to resource patches on the dance floor. Through the feedback and over-representation of profitable resources on the dance floor, recruits will converge to visiting the most profitable resource in vicinity of the hive. The distribution of waggle run durations is the superposition of scouting and recruiting trips.

The distance after which a resource is first discovered by a scout is assumed to follow an exponential distribution (given by $\lambda e^{-\lambda x}$), which is the distribution of the distances to the first object encountered over a linear path when objects are randomly placed. Through the feedback mechanism that the dance floor provides, the colony can, collectively, locate the most profitable resource in its environment. For randomly placed resources in a two-dimensional environment the distance to the nearest point is distributed according to a Rayleigh distribution (given by $2\lambda\pi x e^{-\pi\lambda x^2}$)[46]. Our simulation model (see above) shows that this indeed describes the distances at which recruits visit resources well. Knowing the distance distributions of scout and recruit trips we then assume that the proportion $p$ of all trips are scout trips. With this information we can specify the distributions of distances on the dance floor (see Supplementary Methods for full details).

We implemented this in a full model that describes the distance distribution of an environment that has $n$ different resource types (See

Supplementary Methods). In the full model the number of parameters increases with $2n$ (each resource needs a parameter for the scout and recruit distribution). Even if the number of resources is low, the model tends to overfit. To facilitate estimation of the parameter $p$ we therefore used a simplified model to estimate the fraction of scout trips, where $m$ represents the lowest duration considered, and here a minimum waggle run duration in the data set.

In the simplified the model we assumed that the number of dances depends weakly on distance and there is a sizable quality differences between resources of a non-negligible size and that there is a sizable intensity of the high-quality resource (See Supplementary Methods for detailed derivation). Foragers on scouting trips are more likely to report larger distances than foragers on recruiting trips. By linearising the function that translates the profitability into the number of waggle dance run for the largest waggle run duration and normalising, we arrive at simplified distribution for waggle run durations for scouting trips:

$$f_s(x) = a_s \left( M_s^{-1} b_s e^{-b_s a_s (x-m)} \left[ 1 - a_s x \right]_+ \right) \qquad (1)$$

where we used the shorthand $x_+ = \max(x, 0)$ The parameter $m$ is the minimum recorded duration, $a_s^{-1}$ is the maximum waggle run duration by scouts, $a_s b_s$ is the intensity of resources found by scouts and $M_s = (1 - a_s m) - b_s^{-1} \left( 1 - e^{-b_s(1-a_s m)} \right)$ is the factor that normalises the distribution.

Recruit trips will be predominantly to high-quality resources. Only if the nearest high-quality patch is very far away will there be a more profitable patch of lesser quality available, and this happens only rarely if the intensity of the best quality resource is sizable. After linearising the function that translates the profitability into the number of waggle dance runs for short waggle run durations and normalising the distribution of waggle run durations reported from recruit trips in the simplified model is:

$$f_r(x) = M_r^{-1} 2\pi a_r^2 b_r x e^{-\pi b_r (a_r x)^2} \left[ 1 - a_r x \right]_+ \qquad (2)$$

where

$$M_r = (1 - a_r m) e^{-\pi b r (a_r m)^2} + \frac{\operatorname{erf}(a_r \sqrt{\pi b_r m}) - \operatorname{erf}(\sqrt{\pi b_r})}{2\sqrt{b_r}}$$

is the normalisation factor, the parameter $a_r$ is the rate with which dances repeats depends on distance for recruit trips and $a_r^2 b_r$ is the intensity of high-quality resources reported by recruited foragers.

The simplified distribution function is

$$P(\underline{x} = x) = p f_s(x) + (1-p) f_r(x), \qquad (3)$$

where $f_s(x)$ Is the distribution of distances that reported from scout trips and $f_r(x)$ the distribution of distances reported from recruit trips. The parameter $p$ is the fraction of scout trips, and consequently, $1-p$ the fraction of recruit trips. The likelihood of the parameters given the data of observed $n$ reported distances $(x_1, \dots x_n)$ is

$$L = \prod_{i=1}^{n} p f_s(x_i) + (1-p) f_r(x_i).$$

We determined the maximum likelihood numerically for model fitting and parameter estimation.

## Data collection
Details of data collection, waggle dance decoding and classification of land-use types can be found in full in the Materials and Methods section of Samuelson et al.[24]. Observation hives were located at apiaries in either central London (UK) or the surrounding agricultural land and were each located at least 2 km apart. Visits took place every fortnight between April

and September 2017. On each visit, two hours of continuous waggle dance data was recorded by training a camcorder onto the dance floor. The footage of the dances was decoded manually following methods in ref. 44. Up to 40 dances were decoded per video.

## Statistics and reproducibility
For each real-world hive from the observed dataset, we fitted the distances indicated in the waggle runs to two versions of our model: an "individual" model, where all forage sites are found through scouting, and a "collective" model, where the proportion of scout trips ($p$) can take on any value between 0 and 1.

All models were fit using maximum likelihood estimation[25] by summation over the logarithm of the simplified distribution function outlined in the methods section: model using uninformative priors. The numerical optimisation routine is written in C++ and uses the Nelder-Mead simplex algorithm[47] implemented in the 'NLopt' library[48] and interfaced to R using 'Rcpp'[49].

The most parsimonious model was identified using the Akaike information criterion and Akaike weights[25,50]. Goodness of fit was assessed using the two-sample Kolmogorov-Smirnov (KS) test[27] and implemented in R using the ks.boot function of the package 'Matching' in R[51]. All analysis code was written in R[52].

To determine whether there is variation in the use of the waggle dance within a landscape type we fitted the model to the data from all hives using the individual and collective models and selected the model with the lowest AIC. We also pooled the data for all hives for a landscape type, fitted the individual and collective model to the pooled data and selected the model with the lowest AIC. The sum of the AIC values of the best models for the individual hives in the agri-rural landscape was 2944, for the pooled data it was 3359. For the urban landscape the summed AIC of the best model for the hives was 2059, for the pooled data it was 2394. The difference in was $\Delta AIC = 415$ and $\Delta AIC = 335$ for, respectively, agri-rural and urban landscapes. Corresponding to Akaike weights of 1 for the model for individual hives and 0.000 for the model for with the pooled data, which had essentially no support[25]. See Supplementary Table 1 for details on AIC values.

For each hive, our modelling process resulted in an estimated proportion of recruit trips, $1-p$, henceforth termed "waggle dance use". This proportion was highly variable within the agri-rural category, and so within this group, we sought to identify land-use variables that covary with waggle dance use. For each site, we performed Partial Least Squares analysis based on proportional coverage within 10 land-use categories within a 2.5 km radius around each hive (Supplementary Table 2, see ref. 24 for full classification methods). Prior to conducing the PLS we removed any sites in for which both individual and collective model fit was poor ($n = 1$ site).

As our estimates of waggle dance use are continuous on the interval $[0, 1]$ we used the R package plsRbeta[53] to conduct the PLS and performed a beta regression on the results using the R package betareg[54]. As the betareg package only works on the open interval $(0, 1)$ the data, $x$, was transformed using the following equation: $(x(n-1) + 0.5)/n$ as outlined in the betareg package documentation. After analysis the data was back-transformed to the original values for the plots in Fig. 5.

To test robustness, we performed jack-knifed resampling by removing each site in turn before re-rerunning the PLS analysis, recoding the loadings for each iteration (see Supplementary Fig. 2 for loadings with each site removed). The PLS loadings for each land-use type are plotted as a box plot in Fig. 5 to show the spread of these variable types. A loading was determined to be significantly correlating with the first principal component if contributed more than its expected variance.

Note that significant correlations with land-use types that represent very small proportions of total land cover were not interpreted further (unmanaged green space; Supplementary Table 2; Fig. 5c. Unmanaged green space was also not supported by jackknife analysis).

## Reporting summary

Further information on research design is available in the Nature Portfolio Reporting Summary linked to this article.

## Data availability

All the data used to generate the results reported in this article can be found at: https://doi.org/10.5281/zenodo.7025590 [55]. Any further details on the results reported in this study, or processing of the data are available from the corresponding author on reasonable request.

## Code availability

All the code used to produce the results in this article can be accessed at: https://doi.org/10.5281/zenodo.7025590 [55]. The R code was written in version 4.2 and the Python code in Python 3.9. Details of the R package used and any specific variables or parameters used are specified in the DESCRIPTION file in the R package on Zenodo[55].

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

## Acknowledgements
The authors thank Glenn Ahearn, Mehmet Akiner, Sharon Bassey, Peter Buckoke, John Chapple, Terry Clare, Luke Dixon, Melvyn Essen, Bill Fisher, Clive Hill, the Hive Honey Shop, the Horniman Museum & Gardens, James Makinson, Mark Patterson, Sarah & Vincent Rapley, Simon Rice, Louisa Roscoe, Barnaby Shaw, Sarah Turner, Yalding Beekeepers' Association and the Zoological Society of London for providing research sites, and in some cases for supporting observation hive management and/or providing hives. We thank Huw Fox, Harriet Hall, Jagpreet Hayre, Will Howes, Liberty John, Hana Montague, Michael Sealy, Lucy Tilly-May, Vicky Tubman and Vivitsha Zala for help with waggle dance decoding. Keith McMahon provided beekeeping support, and Lawrence Watson provided the drone. This work was supported through funding by the Biotechnology and Biological Sciences Research Council (BBSRC) through grant BB/M011178/1.

## Author contributions
Designed study: J.P., R.G., E.L., V.J. Performed research: J.P., A.S., E.L., contributed new reagents or analytic tools: J.P., V.J., E.L. Analysed data: J.P., A.S. Wrote the paper: Original Draft: J.P., V.J., E.L. Produced figures: J.P., V.J. Writing - review and editing: J.P., R.G., E.L., V.J. Supervision: R.G., E.L., V.J.

## Competing interests
The authors declare no competing interests.
