## [Transparent Peer Review file · Communications Biology]

Foraging distance distributions reveal how honeybee waggle dance recruitment varies with landscape

Corresponding Author: Professor Vincent Jansen

Version 0:

Reviewer comments:

Reviewer #2

(Remarks to the Author)

In "Honeybees vary communication and collective decision making across landscapes", Palmer et al. (if I understood it correctly) begin by simulating honey bee foraging with and without recruitment to generate two distributions, each showing cumulative frequencies of foraging distances. Then these theoretical distributions are compared to the distributions from a preexisting data set from 20 observation colonies. For 16/20 of the colonies, the measured (from dance decoding), cumulative distance distributions matched the theoretical distribution when recruitment is used. For 4/20 of the colonies measured, the cumulative distance distribution matched the theoretical distribution when recruitment is not used. The authors then looked for, among other things, land use characteristics that might correlate with these 16 colonies. Ultimately they hope that they could build up a picture of when the dance is and is not used by colonies.

Let me begin by saying that went into this review assignment very excited because waggle dance / landscape studies are some of my fav to read. However, I do not think this paper is ready to be submitted. I found it extremely difficult to follow and hard to read.

For example, if my summary above is correct about what is done, then the work is absolutely not captured by the abstract. It's not clear if it is a modelling or experimental or empirical paper. It's not clear what are the take homes (other than variation in the use of the waggle dance, which we knew). There's no detail about what was done (what does inferential method mean?). There are inconsistencies (if I am understanding it correctly? You say in line 29 that it is across landscapes, but in line 31, you say it is within landscapes). There needs to be more specifics, more details, clearer take-home spelled out in the abstract.

Smaller comments

1. Ln 29 abstract - is it really a dichotomy where the bees either use inflexibly or regulated?
2. Ln 64 - beneficial better than justified
3. Ln 75 - didn't you just say that it was used flexibly? why then do you say it is not known if it is used flexibly?
4. Ln 140 - is there any dispute in the assumption that food distribution is unbiased with respect to distance from hive? Would it not be the case that nearby resources might become depleted quicker?
5. Figure 4 - how many hives go into C and D? Is it 4 hives with recruitment not used? Are these distributions made from unequal sample sizes? Or are these just from representative hives?
6. Figure 5 - I have no idea what the take-home is supposed to be from these data. Please be explicit. What am I looking for?
7. Ln 266 - Is this really empirical evidence? Seems like most of your paper is modelling results?
8. Ln 324 - the waggle dance is used less in arable lands, correct? (although I don't know what figure to look at to know this). Shouldn't this be highlighted in the abstract?

In short, I think the approach here is very interesting, and I can see that some interesting take-homes could be made from this work; however, as it is written now, it will be not interesting to a broad readership because it is so hard to make sense of what is going on. The writing is complicated and hard to decipher, and the abstract is unhelpful in distilling the work.

Reviewer #3

(Remarks to the Author)

In this manuscript, Palmer et al. describe modeling of worker bee resource scouting and recruitment via waggle dances. The work presented includes both observations of bee behavior in real hives and analysis of simulated data and model

predictions. The behaviors in question are considered with respect to types of landscapes that differ in their degree of urbanization vs. agricultural usage. The authors describe their main result as the existence of considerable variation in the use of the waggle dance within individual landscape types, as opposed to strictly varying according to landscape type. This is also discussed in the context of the evolutionary history of Apis.

In my opinion the modeling performed appears reasonable but the manuscript would benefit from clearer and more explicit descriptions of motivations, methods, main results, and significance. I am not sure whether the current presentation is of broad enough interest to be published in this journal, but I do think that it could be rewritten to have much broader significance (and to be intelligible) for a variety of readers. I also think that this manuscript would be improved by including more explicit discussion of context in terms of other similar studies. My detailed comments are listed below:

Line 29: "Entirely" is a typo.

Lines 89-96: I think you need to more clearly explain the concept of the "imprint" that you refer to throughout the manuscript, especially since this is a main result.

Line 115: I think it would help orient the reader if you were to briefly mention that you both observed real bees and employed simulations to obtain your results, before delving into this initial section detailing specific results related to simulations.

Figure 2: Explain the exact correspondence of circle color to resource quality.

Figure 3: State in the figure caption whether this is simulated data or real observations.

Figure 3C: Explain the correspondence of the circle color to resource quality. Here and above it might help the reader to put a legend for this information directly in the figure.

Figure 3E: It is not immediately obvious that the outlined yellow/green circle corresponds to one of the aforementioned resource quality circles because of the change in size. Please make this clearer either through further illustration or by explaining it in the caption.

Figure 4: Please add a label distinguishing C vs D so that the figure can be read without having to refer to the caption.

Lines 266-270 & 275-281: These sections are unclear and they should explain the main results. If there is no consistent pattern of influence on waggle dance recruitment from landscape type, then why do you say that "Our results show that honeybees use communication through the waggle dance to different degrees in different environments"? If you found considerable variation among colonies, or according to local patch characteristics, then why is that result not highlighted more? This could relate to genetic or "personality" differences among colonies. You should focus on identifying and clearly explaining what factor has the strongest signal in these results (e.g. landscape type vs. colony identity vs. local patch variation), or more clearly explain contributions from multiple factors or interactions.

Lines 278-281: Is it known or have you shown that such flexible variation is adaptive or optimal based on landscape type? Is it instead possible that some of this results from other differences among colonies, such as colony health, age, or genetics? Or if the main signal is coming from local patch variation, this should be emphasized earlier and more clearly.

Line 325: Is it not possible for you to make broad temporal categories to see if there is any difference? If you are not seeing landscape type consistently driving your results then this might be worth looking into.

Line 337: There is an extra set of parentheses.

Line 354: There is information missing about these recordings. What is the video length? How many bees dance in each recording? If this is pre-existing data set, was it published?

Line 367: Does each resource represent a single flower or a patch? I assume it represents a patch but I can't tell.

Line 370: You should state whether there is any precedent for this type of search path in prior studies (if so, cite them) or if this is original (if so, explain why this would resemble what bees do in nature). This is done well in line 32 with respect to a different part of the simulation.

Line 386: Does this have any sort of timescale or duration built in? Because some nectar can be regenerated, given enough time, but also bee foraging varies over the course of the day. How many visits are required to deplete a resource and does this vary between resources?

Line 411: The distance after which the resource is discovered by bees who watched the dance? Or just after which scouts find it initially in the environment?

Line 412: Why do you think that this distribution specifically describes it well?

Lines 430-431: Is this just a feature of your model or is there evidence for this being the case in reality?

Line 451: It would be easier for the reader if you could restate the definitions of the parameters next to this final version somewhere.

Lines 459-460: How did you open or modify the hive to allow this?

Line 465: So the other, non-scout trips are recruited trips? If not, what are they? Does this mean that in the individual model there is no recruiting?

Line 484: Was this expected? Can you interpret this more somewhere?

Line 511: Should you name the keepers or their farms?

Author Rebuttal letter:

Response to reviewers' comments and list of changes

Reviewer #2 (Remarks to the Author):

In "Honeybees vary communication and collective decision making across landscapes", Palmer et al. (if I understood it correctly) begin by simulating honey bee foraging with and without recruitment to generate two distributions, each showing cumulative frequencies of foraging distances. Then these theoretical distributions are compared to the distributions from a preexisting data set from 20 observation colonies. For 16/20 of the colonies, the measured (from dance decoding), cumulative distance distributions matched the theoretical distribution when recruitment is used. For 4/20 of the colonies measured, the cumulative distance distribution matched the theoretical distribution when recruitment is not used. The authors then looked for, among other things, land use characteristics that might correlate with these 16 colonies. Ultimately they hope that they could build up a picture of when the dance is and is not used by colonies.

Let me begin by saying that I went into this review assignment very excited because waggle dance / landscape studies are some of my fav to read. However, I do not think this paper is ready to be submitted. I found it extremely difficult to follow and hard to read.

We are pleased to hear about your initial excitement and are sorry our manuscript did not live up to your expectations. We have tried to do better and have rewritten aspects of almost every section of the paper in order to improve clarity. Reading your feedback, we realised that we do not seem to have got across the most exciting part of our message very well: that by just observing the waggle dance on the dancefloor it is possible to infer how a colony regulates its collective foraging in response to its environment and landscape structure.

We do this, along the lines as suggested by the reviewer:

1. We first observe that the use of the waggle dance by the colony leaves a statistical imprint on the distribution of the distances (supported by simulation results). The shape of the distribution depends on how much the waggle dance is used.
2. Knowing that the waggle dance influences the shape of the distribution, we derived probability distribution functions for waggle dance use under individual and collective foraging
3. Using maximum likelihood methods we then fit these distributions to observation data from 20 hives and determine which model gives the best fit, and if the waggle dance is used, to what extent it is used. With this technique we can infer the waggle dance use. The fit to the data we achieve in doing so is exceptional. This is an indication that our description is indeed very good in capturing the behaviour of naturally foraging beehives.
4. Finally, we identified factors in natural landscapes that are strong

determinants of waggle dance use.

We have entirely rewritten the abstract (see below). We have also signposted these steps much more clearly in the paper, in the hope that the rationale is easier to follow. We added a roadmap to the results in the last paragraph of the introduction (lines 83-96) and structured the results section by adding the subsections: "The distribution of foraging site distances differs between scout and recruit trips", "Recruitment creates a characteristic pattern in the distribution of foraging distances", "The model captures real-world foraging distributions", "Waggle dance use reflects landscape structure". Please note that the modifications that we have made to increase clarity are too numerous for all individual line numbers to be listed.

For example, if my summary above is correct about what is done, then the work is absolutely not captured by the abstract.

We have rewritten the abstract as suggested by the reviewer to make the main message clearer. The abstract now reads:

"Honeybee (*Apis mellifera*) colonies use a unique collective foraging system, the waggle dance, to communicate and process the location of resources. Here, we present a means to quantify the impact of this system on colony foraging decisions by simply observing the waggle dance on the dancefloor. Through a theoretical model, we show that recruitment leaves a characteristic imprint on the distance distribution of foraging sites that a colony visits, which varies according to the proportion of trips driven by individual search. When we fit this model to the real-world empirical distance distribution of forage sites visited by 20 honeybee colonies in urban and rural landscapes across SE England, obtained via dance decoding, we find considerable variation in the use of dancing in colony foraging, particularly in agri-rural landscapes. In our dataset, reliance on dancing increases as arable land gives way to built-up areas, suggesting that dancing may have the greatest impact on colony foraging in the complex and heterogeneous landscapes of forage-rich urban areas. Our model provides a tool to assess the relevance of this extraordinary behaviour across modern anthropogenic landscape types."

It's not clear if it is a modelling or experimental or empirical paper.

In this paper we develop a statistical method to infer if, and to what degree, waggle dance information is used in bee colonies. Although we use models to find a statistical description of the distributions (which is the basis for any statistical method), but this is not just a modelling paper: we apply our method to analyse empirical data and interpret these results. The paper has both theoretical and empirical aspects.

It's not clear what are the take homes (other than variation in the use of the waggle dance, which we knew).

The take home message is that foraging distance distributions reveal how honeybee waggle dance recruitment varies with landscape: we can infer how much waggle dance information is used in honeybee foraging by observing waggle dances. We establish the degree to which waggle dance information is used is variable and depends on factors in the landscape. Dancing has the greatest impact on colony foraging in forage-rich urban areas.

We knew that individual dance followers do not always respond to spatial information, and we knew something about the circumstances in which they do respond (typically, when their current known resource has reduced in value and they are seeking a new site). Our study aimed to provide a means to identify the landscapes in which that type of situation commonly occurs, and hence, landscapes in which the waggle dance has a significant impact on colony foraging. We have now made this much clearer; for example, at lines 64-81:

"Dancing is a universal feature of honeybee behaviour and is commonly observed in all *Apis mellifera* colonies. However, individual bees only respond to the spatial directions provided by dances under particular circumstances (8,9). For example, foragers that have current knowledge of a resource site are rarely influenced by dances for alternative sites (10), even if those alternatives are more rewarding (11). In contrast, dances are highly influential for foragers whose known sites become depleted (8,12,13), or after temporal gaps in foraging. As a result, the importance of dance communication—and therefore collective decision-making—for a colony's choice of forage sites should depend on resource distribution in the landscape. Environments where recruitment is influential are intriguing, because they are likely to have been important in driving the evolution of the waggle dance, yet empirical attempts to identify them have produced mixed results (14–21).

Initial work, in which dances were rendered meaningless by preventing bees from referencing the sun's position (Figure 1), tentatively linked the benefits of collective foraging to landscape characteristics such as heterogeneity (21). However, empirical attempts to systematically test this relationship have failed to provide support (14,15), and dance disruption has sometimes even been associated with higher, rather than lower, foraging success (19). Consequently, no clear pattern has yet emerged with respect to the ecological conditions that determine whether colonies forage collectively, or as a group of individuals (22)."

We have changed the title of the paper to make the take home message more prominent. It is now: "Foraging distance distributions reveal how honeybee waggle dance recruitment varies with landscape"

There's no detail about what was done

We have added a clearer description of how our method works in the main text at the beginning of the results section. To help with clarity, we focus on the theoretical model, and now explain that the simulation is simply a means to validate our choice of scout and recruit distributions (lines 106-112). Full details on how the simulation model was set up and how our method works is given in the `Material and Methods` section and in the Supplementary Material. We have provided a more intuitive explanation in Fig. 3.

,,, (what does inferential method mean?).

An inferential statistical method leads to an inference about underlying processes. Here we used Likelihood-based inference to establish if, and to what degree, waggle dance information is used in bee colonies from foraging distance distributions. Here, we employed model selection for the inference, which is a powerful statistical paradigm that is increasingly used in ecology (see Burnham and Anderson, 2002)

There are inconsistencies (if I am understanding it correctly? You say in line 29 that it is across landscapes, but in line 31, you say it is within landscapes). There needs to be more specifics, more details, clearer take-home spelled out in the abstract.

We have rewritten the abstract and the title to clarify and to make the take home message clearer.

Smaller comments

1. Ln 29 abstract - is it really a dichotomy where the bees either use inflexibly or regulated?

This phrasing has disappeared owing to the rewriting of the abstract

2. Ln 64 - beneficial better than justified

This phrasing has disappeared owing to the rewriting of the abstract

3. Ln 75 - didn't you just say that it was used flexibly? why then do you say it is not known if it is used flexibly?

This phrasing has disappeared owing to the rewriting of the abstract

4. Ln 140 - is there any dispute in the assumption that food distribution is unbiased with respect to distance from hive? Would it not be the case that nearby resources might become depleted quicker?

The resource availability is not fixed but varies over time as different plants come into flower. In our simulation model we included resource depletion (l. 370-372), as well as new flowers coming available, and this had little to no effect on the shape of the distributions.

What this sentence is meaning to say is that we assume that the positioning of resources is unbiased with respect to the location of the hive: we assume that flowers are distributed over the landscape irrespective of the position of the

hive. We adjusted the wording to reflect this better (line 112-116).

5. Figure 4 - how many hives go into C and D? Is it 4 hives with recruitment not used? Are these distributions made from unequal sample sizes? Or are these just from representative hives?

We analysed the data per hive. Panels (C) and (D) show representative examples of the dance duration distributions for two hives, one without measurable recruitment (C) and one with recruitment (D) which has the characteristic hump. Sample sizes differed between hives, the two hives shown have representative sample sizes (111 and 163). The full set of distributions for all hives is in the supplementary material.

6. Figure 5 - I have no idea what the take-home is supposed to be from these data. Please be explicit. What am I looking for?

The figure shows that there is considerable variation in the amount that the waggle dance information is used (panel a) and that this variation correlates to features of the landscape. We have expanded the caption and description in the text to make the figure easier to follow and interpret (lines 225-250).

7. Ln 266 - Is this really empirical evidence? Seems like most of your paper is modelling results?

All the results in figures 4 and 5 are from empirical data. The paper explains a method and applies this method to the data shown and this allows us to infer the degree to which waggle dance information is used. So yes, this is empirical evidence. We hope that the rewritten introduction and abstract has made this clearer.

8. Ln 324 - the waggle dance is used less in arable lands, correct? (although I don't know what figure to look at to know this). Shouldn't this be highlighted in the abstract?

Thank you for this suggestion. We have made this clearer in the abstract, and have given this more prominence

This is easiest seen in Fig 5b. Going from left to right the information provided by the waggle dance is used more as arable land (coloured green in the inset maps) gives way to build up areas (coloured red in the maps). The axis on this figure is the first principal component from a land-use analysis of all our agricultural sites, which took into account ten land-use categories (e.g. woodland, arable land, etc- given in Fig 5c and in the Supplementary Information). This axis captured an arable-urban gradient, evidenced by the fact that arable land loaded negatively onto it, and urban land loaded positively (figure 5C).

In short, I think the approach here is very interesting, and I can see that some interesting take-homes could be made from this work; however, as it is written now, it will be not interesting to a broad readership because it is so hard to make sense of what is going on. The writing is complicated and hard to decipher, and the abstract is unhelpful in distilling the work.

Thank you for your comments. We have revised the manuscript accordingly and think that it has greatly improved the manuscript.

Reviewer #3 (Remarks to the Author):

In this manuscript, Palmer et al. describe modeling of worker bee resource scouting and recruitment via waggle dances. The work presented includes both observations of bee behavior in real hives and analysis of simulated data and model predictions. The behaviors in question are considered with respect to types of landscapes that differ in their degree of urbanization vs. agricultural usage. The authors describe their main result as the existence of considerable

variation in the use of the waggle dance within individual landscape types, as opposed to strictly varying according to landscape type. This is also discussed in the context of the evolutionary history of Apis.

We apologise that the presentation of our paper was unclear and led to this misunderstanding. We hope that our revised abstract, introduction and results section make it clearer that our study starts by developing a model to describe how colony foraging patterns (i.e. the distribution of distances of foraging sites, relative to the hive) should appear under varying levels of recruitment. We then fit this model to real-world data, to estimate the extent to which foraging is driven by recruitment in each hive. We then relate this quantitative measure of reliance on recruitment to local land-use.

We have made the structure of the paper clearer by extensively rewriting the abstract, introduction, results and discussion. The line numbers for changes are too numerous to list, but we hope that the roadmap included at the end of the introduction (lines 83-96) is particularly useful.

In my opinion the modeling performed appears reasonable but the manuscript would benefit from clearer and more explicit descriptions of motivations, methods, main results, and significance. I am not sure whether the current presentation is of broad enough interest to be published in this journal, but I do think that it could be rewritten to have much broader significance (and to be intelligible) for a variety of readers. I also think that this manuscript would be improved by including more explicit discussion of context in terms of other similar studies.

Thank you for your positive assessment. We have revised the paper to make the motivation, methodology, results, and significance and significance clearer. We have signposted the structure, and have rewritten parts of the abstract, introduction, discussion and the title to address your concerns, as described in the point above.

We are not aware of similar methods that can do this. As requested by the journal, we avoid hyperbolic language in the paper but this is a novel methodology and we are the first to do this. We hope that our rewriting of the manuscript now captures our main message and makes the significance clearer for a general audience. For example, we have highlighted our main take home message: that (a) the extent to which a colony uses recruitment can be inferred simply by observing the dances on the dancefloor, and (b) this can be used to identify landscapes where bees use recruitment. Please see lines 26-28, 64-82, and 84-88 for examples.

My detailed comments are listed below:

Line 29: "Entirely" is a typo.
Removed

Lines 89-96: I think you need to more clearly explain the concept of the "imprint" that you refer to throughout the manuscript, especially since this is a main result.

The nature of the imprint, or hump, is illustrated in Fig 3. We have also inserted the following text to address this:

"To predict the distribution of foraging distances that should be observed across whole colonies, we combined the scout and recruit distributions proportions in a mathematical model (Figure 3; Material and Methods), where the proportion of scouts (α) could vary. Note that the lower the value of α , the higher the number of trips that are driven by recruitment, and thus the more important the role of dance communication for colony foraging. A mixture of scout and recruit trips results in an imprint in the form of a shoulder or "hump" for shorter distances in the cumulative distribution (Fig. 3 H, J). The more recruitment takes place, the more pronounced this hump becomes. The full characterisation of the model can be found in Materials and Methods, eqns. (1) and (2)."

Line 115: I think it would help orient the reader if you were to briefly mention that you both observed real bees and employed simulations to obtain your results, before delving into this initial section detailing specific results related to

simulations.

We have inserted the following paragraph at the end of the Introduction section: results section. "We first develop a theoretical model to establish how colony foraging patterns should appear when recruitment is used to varying degrees, compared to cases where all bees search for food individually. We find that recruitment leaves a characteristic humped imprint on the cumulative distribution of distances reported on the dance floor, the magnitude of which correlates with the use of the waggle dance for collective foraging (see Results). We then fit these theoretical distributions to an empirical data set consisting of observations of waggle dances from 20 real-world hives in two different landscape types—urban and agri-rural—to quantify the relative contribution of waggle dance recruitment to colony foraging decisions in each case. Finally, we relate the variation in waggle dance use that we identify to local land-use patterns."

Figure 2: Explain the exact correspondence of circle color to resource quality.

Done

Figure 3: State in the figure caption whether this is simulated data or real observations.

Done

Figure 3C: Explain the correspondence of the circle color to resource quality. Here and above it might help the reader to put a legend for this information directly in the figure.

Now explained in the caption

Figure 3E: It is not immediately obvious that the outlined yellow/green circle corresponds to one of the aforementioned resource quality circles because of the change in size. Please make this clearer either through further illustration or by explaining it in the caption.

Now explained in caption

Figure 4: Please add a label distinguishing C vs D so that the figure can be read without having to refer to the caption.

Labels added. Thank you, good suggestion.

Lines 266-270 & 275-281: These sections are unclear and they should explain the main results. If there is no consistent pattern of influence on waggle dance recruitment from landscape type, then why do you say that "Our results show that honeybees use communication through the waggle dance to different degrees in different environments"? If you found considerable variation among colonies, or according to local patch characteristics, then why is that result not highlighted more? This could relate to genetic or "personality" differences among colonies. You should focus on identifying and clearly explaining what factor has the strongest signal in these results (e.g. landscape type vs. colony identity vs. local patch variation), or more clearly explain contributions from multiple factors or interactions.

We have entirely rewritten our discussion to address this, at lines 268-304.

"Our results show that there is considerable variation in the impact that waggle dance recruitment has on the distribution of forage sites chosen by honeybee colonies foraging naturally across anthropogenic landscapes (8,9). By simply observing the dancefloor, without any manipulation, it is possible to see that the foraging patterns of some colonies bear a clear hallmark of recruitment, whereas others do not.

What brings about this variation? A "scout" trip is a visit to a site that was originally found by that individual bee through individual search, while a "recruit" trip is a trip to a site that was originally found by that bee by following a dance. In either scenario, once the site has been discovered, a bee typically returns to it repeatedly while it continues to offer reward, irrespective of whether she follows dances for other sites in the meantime within the hive (28). However, most flower species

only offer rewards at certain times of day, and this window may be shortened further by exploitation by other pollinators, so at some point the pay-offs of any known site will decrease, and the bee must find an alternative. At this point, the choice to explore independently or to seek out dance information may depend to some extent on individual intrinsic motivational factors; for example, previous work has found that certain gene expression profiles are predictive of scouting behaviour (32). However, this choice is clearly constrained by the availability of dances indicating alternative foraging sites. If the majority of the colony have been focussing on one single site, or if alternatives are simply not available, scouting is the only available option for forage site discovery. Therefore, we expect that the diversity of forage sites, and forage types, available in the surrounding landscape is likely to be a major factor underlying the extent to which a colony relies on recruitment. The more likely a bee is to encounter dances for alternative sites when newly unemployed, the greater the potential for recruitment to shape colony foraging patterns.

Accordingly, we found patterns in the importance of waggle dance recruitment that reflected landscape structure, such that variation was higher in agri-rural than urban environments, and within the agri-rural category, increased along an axis that reflected a transition from forage-poor agricultural land towards increasing residential development. Agricultural land in the UK is typically considered nutritionally poor for bees, with large areas of limited food availability punctuated by brief availability of rich mass-flowering crops in some areas, while more urbanized residential areas that contain gardens are relatively forage rich, with many diverse small patches of flowers in residential gardens and allotments (29,33,34). We speculate that the dancefloors of hives in these urbanized areas may contain dances advertising multiple alternative sites, such that when rewards decrease at one site, dances advertising others are quickly encountered. At sparser agri-rural sites dominated by arable land, where food is potentially more challenging to find, representation of multiple alternatives on the dancefloor may be rarer. “

Lines 278-281: Is it known or have you shown that such flexible variation is adaptive or optimal based on landscape type? Is it instead possible that some of this results from other differences among colonies, such as colony health, age, or genetics? Or if the main signal is coming from local patch variation, this should be emphasized earlier and more clearly.

We aim to address this question in the future, but with the data we have, it is not possible to answer this question. However, there is no reason to believe that the colonies were significantly different and therefore the main signal is likely to come from variation in the environment. We have removed some of the references to adaptation or selection to make this clearer.

Line 325: Is it not possible for you to make broad temporal categories to see if there is any difference? If you are not seeing landscape type consistently driving your results then this might be worth looking into.

This would require information about the changes in landscape at the time the waggle dances were recorded. This is an interesting suggestion, but we do not have this information because the land-use profiles are derived from satellite maps/drone data collected at a single time point, and so the current data set cannot be used for this. However, we highlight this as a key area for development (lines 304-309):

“Our current dataset lacks the temporal resolution to explore whether recruitment is more common at times when greater diversity is likely within such landscapes, such as late spring, but further work could focus on rural locations systematically chosen to represent a range of floral diversity and abundance (e.g. (35)), with the temporal resolution to focus on specific periods of the year, to identify those ecological contexts in which dance communication has a detectable impact on colony foraging. “

Line 337: There is an extra set of parentheses.

Removed

Line 354: There is information missing about these recordings. What is the video length? How many bees dance in each recording? If this is pre-existing data set, was it published?

We have included this in the material and methods section

Observation hives were located at apiaries in either central London (UK) or the surrounding agricultural land and were each located at least 2km apart. Visits took place every fortnight between April and September 2017. On each visit, two hours of continuous waggle dance data was recorded by training a camcorder onto the dance floor. The footage of the dances was decoded manually following methods in (45).

The details and results of the data has published in reference (24): A. E. Samuelson, R. Schürch, E. Leadbeater. *J. Appl. Ecol.* (2021) <https://doi.org/10.1111/1365-2664.14011>.

Line 367: Does each resource represent a single flower or a patch? I assume it represents a patch but I can't tell.

In our simulation models these are objects that the foragers discover. As honeybees waggle dances are unlikely to have an accuracy down to a single plant, we interpret this as a patch. To address your point, we now also used this wording in the simulation model description.

Line 370: You should state whether there is any precedent for this type of search path in prior studies (if so, cite them) or if this is original (if so, explain why this would resemble what bees do in nature). This is done well in line 32 with respect to a different part of the simulation.

There is evidence that search path are random as described. We have added the line.

Foraging bees follow independent flight segments with random length and direction (40) [<https://doi.org/10.1242/jeb.009563>].

Line 386: Does this have any sort of timescale or duration built in? Because some nectar can be regenerated, given enough time, but also bee foraging varies over the course of the day. How many visits are required to deplete a resource and does this vary between resources?

Resources have a varying quality and are periodically replaced in the simulation model with new resource of a random quality on about once in simulation; the results are not very sensitive to rate of replacement. We have now indicated this in the manuscript (lines 352-354)

Line 411: The distance after which the resource is discovered by bees who watched the dance? Or just after which scouts find it initially in the environment?

It is the distribution of a scout discovering the resource at first. Sentence changed to

The distance after which a resource is first discovered by a scout is assumed to follow an exponential distribution (given by $P(x) = \lambda e^{-\lambda x}$), which is the distribution of the distances to the first object encountered over a linear path when objects are randomly placed.

Line 412: Why do you think that this distribution specifically describes it well?

That is the distribution of distances to the first object encountered along a linear path, where objects are randomly placed in the environment (this is a basic result from the theory of Poisson processes). Sentence changed as above to make this clearer.

Lines 430-431: Is this just a feature of your model or is there evidence for this being the case in reality?

The extraordinary fit of the data to our model shows that the model describes what is happening in reality with accuracy (see fits in Fig 4 and supplementary material)

Line 451: It would be easier for the reader if you could restate the definitions of

the parameters next to this final version somewhere.

We have added:

.. where φ (ρ) is the distribution of distances that reported from scout trips, ρ (ρ) the distribution of distances reported from recruit trips. The parameter ρ is the fraction of scout trips, and consequently, $1-\rho$ the fraction of recruit trips.

Lines 459-460: How did you open or modify the hive to allow this?

Observation hives were used (l. 448). They have a glass panel through which the dance floor can be observed

Line 465: So the other, non-scout trips are recruited trips? If not, what are they? Does this mean that in the individual model there is no recruiting?

Correct. In the individual model there is no collective foraging and all trips are scout trips.

Line 484: Was this expected? Can you interpret this more somewhere?

We are describing the method here. A $\Delta AIC=415$ and $\Delta AIC=335$ means that one model essentially has no support (see (25)). The magnitude of the ΔAIC we did not anticipate, but is a reflection of how well the model describes that data. The results are discussed in the main text in detail and we emphasize the quality of the fit in other places in the text (l. 194).

Line 511: Should you name the keepers or their farms?

The colonies were managed by the research team. They were placed on farms but the farmers had no involvement in the management of the hives.

However, we have previously acknowledged their goodwill when publishing the original dataset, and for consistency now do so again here. Thank you for the suggestion.

Version 1:

Reviewer comments:

Reviewer #3

(Remarks to the Author)

I have previously reviewed this manuscript. In my previous review, while I found the work significant and very detailed, I struggled with the clarity of many explanations and the overall framing. In my opinion this new draft is greatly improved. I am satisfied with their response to my prior comments. I still have some new comments and suggested changes for this draft, but the authors have succeeded in emphasizing the main points of their work in this revision.

Line 28: In order to quantify the impact of dancing on colony foraging decisions, wouldn't you also need to know what the colony actually ends up doing in addition to observing the dance? I don't really understand this statement.

Line 33: Are you allowed to use an abbreviation in an abstract?

Line 37: This is much better explained now.

Line 90: I think this needs to be described in slightly more detail (or maybe just enclosed in quotes) the first time it is mentioned.

Line 114: There is a typo or mis-formatted citation here.

Figure 2: Are panels A and B individual representative runs from the simulation? Also, while I appreciate the added details about the meaning of colors, disks, etc. in the caption, it would be much better for the reader if you could inset keys/legends directly in the figure.

Line 142: This is a great description of what you mean about the "hump", so maybe you could move this information to the first time the concept is mentioned.

Figure 3: For the y axis on the plots in J and H, is this also a log scale? If so it should be specified.

Figures 4 & 5: It would be better for the reader in terms of understanding these figures if you could inset the additional information about the meaning of colors and shapes (for figure 4) directly in the corresponding figure panels.

Line 271: What does this mean? Does this mean that you can see individual bees being recruited because of how they are moving? Or do you mean that if you compare the dancing to the colony's actual foraging patterns, some instances indicate that bees were recruited and others don't?

Line 289: Is there evidence for this? I follow the logic in this paragraph up to this point, but I am confused about why a newly unemployed bee could not be recruited to a previously known site if it is being advertised by current dancing. Wouldn't being recruited to the same site as a previous trip (i.e. not an alternative site) still count as being recruited if there are still workers indicating that site via dancing? Or do they stop advertising sites once some bees already know about them/have visited them?

Line 458: I would suggest explaining this difference in the collective vs. the individual model in this level of detail the first time mention it earlier in the paper. It is explained very clearly here and makes sense but I didn't exactly have this in mind when I first read the names of these models.

Author Rebuttal letter:

Response to reviewer's comments and list of changes

Reviewers' comments:

Reviewer #3 (Remarks to the Author):

I have previously reviewed this manuscript. In my previous review, while I found the work significant and very detailed, I struggled with the clarity of many explanations and the overall framing. In my opinion this new draft is greatly improved. I am satisfied with their response to my prior comments. I still have some new comments and suggested changes for this draft, but the authors have succeeded in emphasizing the main points of their work in this revision.

Line 28: In order to quantify the impact of dancing on colony foraging decisions, wouldn't you also need to know what the colony actually ends up doing in addition to observing the dance? I don't really understand this statement.

Thank you for highlighting that this is confusing. We used this phrasing because we do know what the colony ends up doing: the distribution of distances decoded from dances tells us where the colony is sending its foragers. We sought to establish how much that observed distribution differs from the distribution that we would have observed if no recruitment had taken place, and also emphasize that our methodology is non-disruptive and pure observational. To make this clearer, we have rephrased (Line 28):
"Here, we present a means to quantify the effect of recruitment on colony forager allocation across the landscape by simply observing the waggle dance on the dancefloor."

Line 33: Are you allowed to use an abbreviation in an abstract?

We have removed the abbreviation and have now spelled this out .

Line 37: This is much better explained now.

Thank you!

Line 90: I think this needs to be described in slightly more detail (or maybe just enclosed in quotes) the first time it is mentioned.

We have expanded the description, but as this is a summary of the results we have left the fuller explanation in the results section. We have added in quotation marks, as, but also now make our bracketed direction to the full description provided later in the manuscript more obvious and precise (line 90).

Line 114: There is a typo or mis-formatted citation here.
Corrected

Figure 2: Are panels A and B individual representative runs from the simulation? Also, while I

appreciate the added details about the meaning of colors, disks, etc. in the caption, it would be much better for the reader if you could inset keys/legends directly in the figure.

We have moved the description of patch quality to the figure as a legend, as suggested. Panels A and B are diagrammatic representations of honeybee foraging and we have adjusted the figure caption to make this clear.

Line 142: This is a great description of what you mean about the "hump", so maybe you could move this information to the first time the concept is mentioned.

Thank you- please see response above.

Figure 3: For the y axis on the plots in J and H, is this also a log scale? If so it should be specified.

We have adjusted the axes labels to make this clear.

Figures 4 & 5: It would be better for the reader in terms of understanding these figures if you could inset the additional information about the meaning of colors and shapes (for figure 4) directly in the corresponding figure panels.

We have added a legend to Figure 5. For Figure 4 it was not feasible to put a legend without disrupting the lay-out of the figure. We have instead added the shapes to the legend to make it easier for the reader.

Line 271: What does this mean? Does this mean that you can see individual bees being recruited because of how they are moving? Or do you mean that if you compare the dancing to the colony's actual foraging patterns, some instances indicate that bees were recruited and others don't?

We have changed our phrasing to make the meaning clearer (line 270):

"We do this by using a purely observational approach that does not disrupt the natural foraging behaviour of the colony: By simply decoding the dances produced by a colony's workforce, and fitting our model to the foraging site distances obtained as a result, it is possible to see that the foraging patterns of some colonies bear a clear hallmark of recruitment, whereas others do not"

Line 289: Is there evidence for this? I follow the logic in this paragraph up to this point, but I am confused about why a newly unemployed bee could not be recruited to a previously known site if it is being advertised by current dancing. Wouldn't being recruited to the same site as a previous trip (i.e. not an alternative site) still count as being recruited if there are still workers indicating that site via dancing? Or do they stop advertising sites once some bees already know about them/have visited them?

Your comment made us realise that the explanation provided in the original manuscript was unclear. In response, we have (a) added evidence that the proportion of bees opting to scout is inversely proportional to the number of dances in the hive (b) simplified our explanation to make it more general and avoid unnecessary detail. The key point is that the fewer the number of dances in the hive, the more limited the potential for an individual to choose the recruit strategy. (However, in answer to your question- bees never stop advertising a site as long as it is still rewarding, but the situation we described referred to a site that had become unrewarding). Thank you for highlighting this, as we feel that the explanation has been improved by your comment (line 274-285):

"What causes this variation? Our metric for assessing the impact of dance recruitment is the proportion of bees that follow a "recruit" strategy, sampling from the dance floor before leaving the hive to find the site communicated by the dance, rather than searching independently. At the proximate level, the factors that underlie the choice to act as scout or recruit are not well understood, although previous work has found that certain gene expression profiles are predictive of scouting behaviour (32) and that the tendency to act as a recruit is greater in younger bees (33,34). But it is clear that these strategies are sufficiently flexible to allow changes in the proportion of scouts with local foraging conditions, and Seeley (35) observed through intensively tracking individual bees in observation hives that the proportion of scouts decreases dramatically with forage availability, as the number of dances increases (36). Thus, there is reason to expect that the hives we identified as relying heavily on dance recruitment are those that are in forage-rich areas. "

Line 458: I would suggest explaining this difference in the collective vs. the individual model in this level of detail the first time mention it earlier in the paper. It is explained very clearly here and makes sense but I didn't exactly have this in mind when I first read the names of

these models.

We have used a similar description as in the methods section to explain the difference in collective vs individual in the main text (l. 167-172):

“For each hive, we explored whether foraging was best described as an individual or collective venture. To do so we fitted two versions of our model using maximum likelihood methods. The first was an “individual” model, where all forage sites are found through scouting, and the second was a “collective” model, where the proportion of scout trips (p) could take on any value between 0 and 1..”

Thank you for your feedback, which has helped us improve the manuscript further.

Version 2:

Reviewer comments:

Reviewer #3

(Remarks to the Author)

I have reviewed this manuscript two times, and I think it can be published and is a good study. I have some remaining opinions about figure captions and the phrasing of some new text that I think can be resolved by the editor without need for a further round of review. The changes to the figures are generally very useful.

Line 91: There is an extra quotation mark.

Figure 2: You explained that these are "diagrammatic", i.e. not illustrating real data or simulation runs (I think?). Adding "A diagram showing..." is somewhat helpful but I don't think it actually communicates what this is. I would change it to something like "hypothetical illustration of a typical foraging bout". If it does correspond to any data or run then I would write something like "representative typical foraging bout".

Figure 4: "For Figure 4 it was not feasible to put a legend without disrupting the lay-out of the figure." - I disagree. There is space in the upper right corner of panels C & D for a small color legend for teal vs. red lines. In panel A a small color/shape legend could be added just to left of the compass rose, or just under the compass rose. The editor can speak to whether the available space is sufficient for adequate text size.

Line 275: I think this added text is helpful. However I cannot follow this phrase: "...sampling from the dance floor before leaving the hive to find the site communicated by the dance, rather than searching independently." Is the subject of the sentence changing part of the way through this phrase? Are you sampling or are the bees sampling? Are you tracking them as they leave the hive or are we imagining the POV of a bee as she leaves the hive?

Author Rebuttal letter:

Response to reviewer's comments and list of changes

Reviewer's comments:

Reviewer #3 (Remarks to the Author):

I have reviewed this manuscript two times, and I think it can be published and is a good study. I have some remaining opinions about figure captions and the phrasing of some new text that I think can be resolved by the editor without need for a further round of review. The changes to the figures are generally very useful.

Line 91: There is an extra quotation mark.

Extra quotation mark removed

Figure 2: You explained that these are "diagrammatic", i.e. not illustrating real data or simulation runs (I think?). Adding "A diagram showing..." is somewhat helpful but I don't think it actually communicates what this is. I would change it to something like "hypothetical illustration of a typical foraging bout". If it does correspond to any data or run then I would write something like "representative typical foraging bout".

Caption changed (line 136) as suggested. Now reads "A hypothetical illustration of a typical foraging honeybee,"

Figure 4: "For Figure 4 it was not feasible to put a legend without disrupting the layout of the figure." - I disagree. There is space in the upper right corner of panels C & D for a small color legend for teal vs. red lines. In panel A a small color/shape legend could be added just to left of the compass rose, or just under the compass rose. The editor can speak to whether the available space is sufficient for adequate text size.

We have changed the figure to include the legend as requested

Line 275: I think this added text is helpful. However I cannot follow this phrase: "...sampling from the dance floor before leaving the hive to find the site communicated by the dance, rather than searching independently." Is the subject of the sentence changing part of the way through this phrase? Are you sampling or are the bees sampling? Are you tracking them as they leave the hive or are we imagining the POV of a bee as she leaves the hive?

We have changed the text (line 291-3) to clarify. The text now reads

"Bees that use this strategy sample a dance from the dance floor before leaving the hive to find the site communicated by the dance, rather than searching independently."

Thank you again for your feedback and are pleased to hear you think it is a good study. Your constructive feedback has helped to improve the manuscript.
